# Multi-Functional MPT Protein as a Therapeutic Agent against *Mycobacterium tuberculosis*

**DOI:** 10.3390/biomedicines9050545

**Published:** 2021-05-13

**Authors:** Jae-Sung Kim, Euni Cho, Seok-Jun Mun, Sojin Kim, Sun-Young Kim, Dong-Gyu Kim, Wooic Son, Hye-In Jeon, Hyo-Keun Kim, Young-Jin Jeong, Sein Jang, Hyun-Sung Kim, Chul-Su Yang

**Affiliations:** 1Department of Bionano Technology, Hanyang University, Seoul 04673, Korea; sung901017@hanyang.ac.kr (J.-S.K.); eunicho@hanyang.ac.kr (E.C.); moon07101@hanyang.ac.kr (S.-J.M.); tjsdud7969@hanyang.ac.kr (S.-Y.K.); 2Institute of Natural Science & Technology, Hanyang University, Ansan 15588, Korea; 3Center for Bionano Intelligence Education and Research, Ansan 15588, Korea; yjs6124@hanyang.ac.kr (W.S.); jhi1007@naver.com (H.-I.J.); wow1560@naver.com (H.-K.K.); youngjin3489@naver.com (Y.-J.J.); jtpdls26@naver.com (S.J.); 4Department of Molecular and Life Science, Hanyang University, Ansan 15588, Korea; thwls2646@hanyang.ac.kr (S.K.); hellohy@hanyang.ac.kr (D.-G.K.); 5Department of Pathology, Hanyang University College of Medicine, Seoul 04673, Korea; hhnt5841@gmail.com

**Keywords:** *Mycobacterium tuberculosis*, MPT peptide, TBK1, p47phox, hexokinase 2, macrophages

## Abstract

*Mycobacterium tuberculosis* (MTB), the causative agent of tuberculosis (TB), avoids the host immune system through its virulence factors. MPT63 and MPT64 are the virulence factors secreted by MTB which regulate host proteins for the survival and proliferation of MTB in the host. Here, we found that MPT63 bound directly with TBK1 and p47phox, whereas MPT64 interacted with TBK1 and HK2. We constructed a MPT63/64-derived multifunctional recombinant protein (rMPT) that was able to interact with TBK1, p47phox, or HK2. rMPT was shown to regulate IFN-β levels and increase inflammation and concentration of reactive oxygen species (ROS), while targeting macrophages and killing MTB, both in vitro and in vivo. Furthermore, the identification of the role of rMPT against MTB was achieved via vaccination in a mouse model. Taken together, we here present rMPT, which, by regulating important immune signaling systems, can be considered an effective vaccine or therapeutic agent against MTB.

## 1. Introduction

Tuberculosis (TB), caused by *Mycobacterium tuberculosis* (MTB), is one of the most important infectious diseases worldwide, due to its high mortality [1]. MTB infects the host macrophages and survives through evading the host immune system. Lung granuloma is a representative lesion in TB [2,3]. MTB, which possesses several immunogenic proteins against the host cells [4], persists inside the granuloma for survival and proliferation, although its function is still not clear.

The function of numerous secretory proteins of MTB is still unknown. MTB’s secretory antigens are essential for interacting with host proteins and regulating the immune response, and for bacterial proliferation and survival [5]. These proteins secreted during infection can determine the pathways of the adaptive immune system, such as activating effector T-cells. Thus, investigating the precise role of the secretory antigens is essential for understanding the pathogenicity of TB [6].

MPT63 and MPT64 are secretory immunodominant antigens of MTB, which are highly expressed in active TB. They can be detected in the macrophages of a TB lesion [7]. In previous studies, MPT63 induced cell death of macrophages through a conformational switch dependent on the pH of the host [8]. Additionally, MPT63 enhanced the phagocytic activity by upregulating the secretion of TNF-α and IL-6 in murine peritoneal macrophages [9]. Located in the RD2 region, MPT64 induced the production of IFN-γ in murine macrophages and patients with TB, and reduced apoptosis by increasing the expression of TGF-β and reducing inflammation [10,11]. MPT63 and MPT64 are both utilized as disease markers for TB and appear to be potential candidates for the development of a vaccine against TB [12,13,14,15,16]. However, the mechanism of interaction between MPT63 or MPT64 with host proteins is still unclear.

The immune signaling pathways in TB begin with pathogen recognition by pattern recognition receptors, such as TLRs, NLRs, CLRs, and scavenger receptor [17]. Increasing the expression of inflammatory cytokines and chemokines, through the activation of the NF-κB signaling pathway, and reactive oxygen species (ROS), is essential to eliminate active MTB in the host [18,19]. Nonetheless, intracellular MTB survives in the host by regulating specific host metabolic pathways. In particular, IFN-β is crucial for the mechanism of immune evasion in TB. While during a viral infection IFN-β protects the host, in MTB infection this protein functions in the opposite way [20]. Cytosolic DNA from MTB induces the cGAS-STING1-TBK1-IFN-β pathway for the suppression of activation of NLRP3 inflammasome, which contributes toward the antimicrobial activity in the host [21,22]. Additionally, MTB inhibits the function of ROS via catalase peroxidase, including KatG, TrxB2, and several antigens of MTB, such as ESAT-6 and CFP-10 [4,23,24].

In this study, we found that MPT63 and MPT64 interacted with several proteins in host macrophages including TBK1, p47phox, and HK2. We examined the domains of interaction with TBK1, p47phox, and HK2 in MPT63 or MPT64. TBK1 and p47phox peptides increased the antimycobacterial effect through overexpression of pro-inflammatory cytokine and ROS, and under-expression of IFN-β. We constructed a recombinant multifunctional MPT protein (rMPT) containing the peptides which interacted with TBK1, p47phox, and HK2. This protein was effective in regulating the burden of MTB by increasing inflammation. Additionally, rMPT increased the mycobactericidal effect against MTB via vaccination. Taken together, rMPT can be assumed to be a potential candidate for a vaccine against TB.

## 2. Materials and Methods

### 2.1. Mice and Cell Culture

Wild-type C57BL/6 mice were purchased from Samtako Bio Korea (Gyeonggi-do, Korea). TBK1^−/^^−^ and p47phox ^−/^^−^ mice were a generous gift from Dr. Chul-Ho Lee (Laboratory Animal Resource Center, Korea Research Institute of Bioscience and Biotechnology, Daejeon, Korea). LysMCre mice (B6.129P2-*Lyz2^tm1(cre)Ifo^*/J) mice were purchased from Jackson laboratories and Hexokinase II (HK2)-floxed mice (C57BL/6 background) mice were a kind gift from Dr. Nissim Hay (University of Illinois, Chicago) [25]. Cre-mediated recombination was confirmed by PCR using genomic DNA from isolated peritoneal macrophages and primers flanking the floxed region, as described previously [25]; the absence of HK2 expression in these macrophages was confirmed by immunoblot analysis using a HK2-specific antibody (Santa Cruz, B-8, Dallas, TX, USA). All mice were housed in a specific pathogen-free (SPF) facility on the basis of standard humane animal husbandry protocols.

Primary bone marrow-derived macrophages (BMDMs) were isolated from C57BL/6 mice and cultured in DMEM for 3–5 d in the presence of M-CSF (416-ML, R&D Systems, Minneapolis, MN, USA), as described previously [26]. HEK293T (ATCC-11268; American Type Culture Collection, Manassas, VA, USA) or THP-1 (ATCC-TIB-202) cells were maintained in DMEM or RPMI1640 (Gibco, Waltham, MA, USA) containing 10% FBS (Gibco), sodium pyruvate, nonessential amino acids, penicillin G (100 IU/mL), and streptomycin (100 μg/mL). Human monocytic THP-1 (ATCC TIB-202) cells were grown in RPMI 1640/glutamax supplemented with 10% FBS and treated with 20 nM PMA (Sigma-Aldrich, St. Louis, MO, USA) for 24 h to induce their differentiation into macrophage-like cells, followed by washing three times with PBS. Transient transfections were performed using calcium phosphate (Clontech, Takara Bio Inc., Shiga, Japan) in 293T, according to the manufacturer’s instructions. THP-1 stable cell lines were generated by transfections performed using Lipofectamine 3000 (Invitrogen, Waltham, MA, USA) and then a standard selection protocol with 400–800 μg/mL of G418.

### 2.2. Bacterial Strains

*E. coli* DH5α and BL21 were grown in flasks using LB medium for genetic manipulations or protein overexpression. Cultures of *M. bovis* BCG and MTB H37Rv (provided by Dr. R. L. Friedman, University of Arizona, Tucson, AZ, USA) were prepared as described previously [27]. The effective concentration of lipopolysaccharide was <50 pg/mL in these experiments, with a bacterium-to-cell ratio of 10:1. For all assays, midlog phase bacteria (absorbance 0.4) were used. Bacterial strains were divided into 1-mL aliquots and stored at −70 °C.

### 2.3. Recombinant Protein 

To obtain MTB H37Rv strain-derived MPT63/64 (GenBank accession no. NP_216442 and NP_216496) recombinant protein, MPT63 amino acid (50–56, 152–158), MPT64 amino acid (24–28, 34–38, 187–193), and R9 seq were cloned with an N-terminal 6xHis tag into the pRSFDuet-1 Vector (Novagen, St. Louis, MO, USA) and induced, harvested, and purified from *Escherichia coli* expression strain BL21(DE3) pLysS, as described previously [27] and in accordance with the standard protocols recommended by Novagen. rMPT was dialyzed with permeable cellulose membrane and tested for lipopolysaccharide contamination with a *Limulus* amebocyte lysate assay (BioWhittaker, Lonza, Basel, Switzerland) and contained <20 pg/mL at the concentrations of rMPT proteins used in the experiments described here.

### 2.4. Antibodies

Abs specific for Flag (D-8), GST(B-14), V5 (E10), p22phox (FL-195), gp91phox (H-60), p47phox (H-195), p67phox (H-300), HK1 (G-1), HK2 (B-8), HK3 (A-9), and Actin (I-19) were purchased from Santa Cruz Biotechnology (Dallas, TX, USA). Specific antibodies against phospho-p47phox (S304, ABIN1526728), (S345, ABIN482777), (S359, ABIN482335), and (S370, ABIN1989372) were purchased from St John’s Laboratory. The antibodies to IRF3 (ab68481), His (ab18184), Myc (ab9106), and AU1 (ab3401) were purchased from Abcam (Cambridge, UK). The antibody to TBK1 (E8I3G) and phospho-TBK1(S172) (D52C2) were from Cell Signaling Technology (Danvers, MA, USA) and STING (NBP 2–24683) was from Novus Biologicals (Centennial, CO, USA). 

### 2.5. Plasmid Construction

The plasmid encoding full-length of the MPT63 (NR-15618) and MPT64 (NR-13273) plasmids were provided by BEI Resources; TBK1 (87443) and HK2 (25529) were purchased by Addgene (Watertown, MA, USA); p47phox were previously described [28]. Plasmids encoding different regions of MPT63, MPT64, TBK1, p47phox, HK2 were generated by PCR amplification from full-length cDNA and subcloning into a pEBG derivative encoding an N-terminal GST epitope tag between the *BamHI* and *NotI* sites. All constructs for transient and stable expression in mammalian cells were derived from the pEBG-GST mammalian fusion vector and the pEF-IRES-Puro expression vector. All constructs were sequenced using an ABI PRISM 377 automatic DNA sequencer to verify 100% correspondence with the original sequence.

### 2.6. Interaction Kinetic Analyses of the MPT63 or MPT64 with Binding Partners

The interactions of MPT63-TBK1 or p47phox and MPT64-TBK1 or HK2 were monitored using a Fluoromax-4 spectrofluorometer (HORIBA Scientific), and were performed as previously described [29]. Briefly, MPT63 or MPT64 was labeled with BODIPY FL Iodoacetamide (ThermoFisher Scientific, Waltham, MA, USA), according to the manufacturer’s instructions. Labelled MPT63 or MPT64 was excited at 350 nm, and detection was through a cutoff filter at 512 nm. Fluorescently labelled MPT63 or MPT64 was titrated with unlabeled TBK1, p47phox, or HK2 for the kinetic analysis. The excitation and emission wavelengths used were 498 mm and 518 nm, respectively. The data obtained were fitted using the Grafit program. All fluorescence measurements were performed at 25 °C in 30 mM Tris, pH 7.4, 150 mM NaCl and 1 mM dithiothreitol.

### 2.7. Enzyme-Linked Immunosorbent Assay

Cell culture supernatants and mice sera were analyzed for cytokine content using a BD OptEIA ELISA set (BD Pharmingen, San Jose, CA, USA) for the detection of TNF-α, IL-6, IL-2, and IFN-γ; Mouse IFN Beta ELISA Kit (PBL Assay Science, Piscataway, NJ, USA) for IFN-β. All assays were performed as recommended by the manufacturer.

### 2.8. Generation of HK2 Knockout Cells

A HK2 KO THP-1 cell line was established by CRISPR/Cas9-mediated genome editing by HK2 Human Gene Knockout Kit (CRISPR) (KN209482) from OriGene Technologies, Inc. (Rockville, MD, USA). HK2 KO cells were selected by puromycin. All assays were performed as recommended by the manufacturer.

### 2.9. MTB Infection In Vitro and In Vivo

For in vitro experiments, cells were infected with MTB for 2–4 h. Then, cells were washed with PBS to remove extracellular bacteria, supplied with fresh medium, and incubated at 37 °C for the indicated time periods. For in vivo experiments, female SPF C57BL/6 mice were 6–8-weeks old during the course of the experiments and were age- and sex-matched in each experiment. No additional randomization or blinding was used to allocate experimental groups. Mice were i.n. injected with MTB (1 × 10^4^ CFU/mouse). After 5 wks, mice were sacrificed for harvesting of the lungs, spleens, and livers. Mice were maintained in biosafety level 3 laboratory facilities. 

### 2.10. Peptides

R9-conjugated MPT peptides were commercially synthesized and purified in acetate salt form to avoid abnormal responses in cell by Peptron (Korea). The endotoxin content, was measured by Limulus amebocyte lysate assay (BioWhittaker) and contained less than 3–5 pg/mL at the concentrations of the peptides used in experiments. 

### 2.11. Histology

For immunohistochemistry of tissue sections, mouse lungs were fixed in 10% formalin and embedded in paraffin. Paraffin sections (4 μm) were cut and stained with hematoxylin and eosin (H&E). Histopathologic score was established on the basis of the numbers and distribution of inflammatory cells and the severity of inflammation within the tissues [30,31] in which a board-certified pathologist independently scored each organ section without prior knowledge of the treatment groups. A histological score ranging from 0–4 was ascribed to each specimen.

### 2.12. In Vivo Imaging

rMPT/Cy5.5 was prepared by adding streptavidin-conjugated Cy5.5 dye to rMPT. rMPT/Cy5.5 were administered into mice via i.n in MTB-infected mice. To study tissue bio-distribution, mice were sacrificed at different time points post-administration and the major organs were excised and imaged using the IVIS Spectrum-CT in vivo imaging system (PerkinElmer, Inc., Waltham, MA, USA).

### 2.13. Miscellaneous Procedures

Details of protein purification and mass spectrometry, GST pulldown, immunoprecipitation, immunoblot assays, confocal fluorescence microscopy, and MTT assay are provided in the Appendix A.

### 2.14. Statistical Analysis

All data were analyzed using Student’s *t*-test with Bonferroni adjustment for multiple comparisons, and are presented as mean ± SD. Statistical analyses were conducted using the SPSS (Version 12.0) statistical software program (SPSS, Chicago, IL, USA). Differences were considered significant at *p* < 0.05. For survival, data were graphed and analyzed by the product limit method of Kaplan and Meier, using the log-rank (Mantele-Cox) test for comparisons using GraphPad Prism (version 5.0, La Jolla, CA, USA).

## 3. Results

### 3.1. MPT63 Directly Interacts with TBK1 and p47phox 

To determine the binding partners of MPT63 in host macrophages, recombinant MPT63 (rMPT63) was subjected to co-immunoprecipitation with THP-1 macrophage cell lysates. First, we constructed the plasmid and purified rMPT63 by using a 6x His bacterially expressed system (Appendix A). The purified rMPT63 complexes were identified by mass spectrometry analysis and included TANK-binding kinase (TBK1, 83K), receptor-interacting serine/threonine-protein kinase 1 (RIP1, 75K), RAC-alpha serine/threonine-protein kinase (AKT, 55K), Neutrophil cytosol factor 1 (p47phox, 45K), and p38 mitogen-activated protein kinase (p38MAPK, 38K) (Figure 1A and Appendix A). As TBK1 and p47phox were related to the expression of IFN-β and to increasing the levels of ROS in TB, we selected TBK1 and p47phox to further study their interaction with MPT63 [20,32]. To investigate the endogenous interaction in macrophages, we treated THP-1 cells with rMPT63 and co-immunoprecipitated. In THP-1 cells, endogenous binding demonstrated that rMPT63 interacted with TBK1 and p47phox and, interestingly, with their phosphorylated forms as well (S345 and S359) (Figure 1B). Additionally, the in vitro interaction between MPT63 and TBK1 or p47phox, measured using a fluorescence binding assay with recombinant proteins and fluorescently labelled MPT63 and TBK1 or p47phox, showed an adequately high affinity (TBK1, 178nM; p47phox, 345nM) (Figure 1C,D). MPT63 contains a signal peptide, N-terminal, middle, and C-terminal domain. To find the domain for interaction between MPT63 and TBK1 or p47phox, we used each of the domains of GST-MPT63 and Flag-TBK1 or V5-p47phox. In 293T cells, the N-terminal of MPT63 bound with TBK1 and 48–56 peptide was essential for the interaction between MPT63 and TBK1. Specifically, 50–56 amino acids, except G53, seemed to be fundamental for binding in MPT63 and TBK1. To examine the binding domain between MPT63 and p47phox, we transfected GST-MPT63 and V5-p47phox in 293T cells, followed by GST pull-down. The C-terminal of MPT63 was associated with p47phox, particularly the 150–159 amino acids in the C-terminal domain of MPT63 were important for binding with p47phox. Additionally, E152, D153, and E158 in MPT63 were key amino acids in the interaction between MPT63 and p47phox (Figure 1G,H). Furthermore, to investigate the binding domain in TBK1 or p47phox to MPT63, various mammalian fusions and truncated mutants of GST-TBK1 or p47phox were pulled down with the wild-type Myc-MPT63. In 293T cell, the kinase domain in TBK1 was essential to bind MPT63, and the PX domain in p47phox was important for the interaction with MPT63 (Appendix A). In summary, these results showed that MPT63 binds to TBK1 and p47phox via the N-terminal or C-terminal in MPT63.

### 3.2. MPT64 Is Directly Associated with TBK1 and HK2

To investigate the binding partner of MPT64 in host proteins, we treated the recombinant MPT64 (rMPT64) and immunoprecipitated it in THP-1 cell lysate. We produced rMPT64 in a bacterially expressed system as MPT63 (Appendix A). In the purified rMPT64 complex, rMPT64 interacted with several host proteins, including hexokinase 2 (HK2, 102K), TBK1 (83K), protein kinase Cα (PKCα, 76K), TNF receptor associated factor 6 (TRAF6, 60K), and p38MAPK (38K) (Figure 2A and Appendix A). Interestingly, TBK1 also bound to MPT64. Furthermore, we focused our next steps on HK2, as this is an essential protein for glycolysis. As reported previously, HK2 is essential for the immune response via regulation of the glycolysis metabolism in TB [33]. We treated rMPT64 to investigate its endogenous binding with TBK1 and HK2. Interestingly, phosphorylated TBK1 (S172) also interacted with rMPT64 (Figure 2B). Moreover, the in vitro binding between MPT64 and TBK1 or HK2 was analyzed in a binding assay using recombinant proteins with fluorescently labelled MPT64 bound with TBK1 or p47phox in high affinity (TBK1, 193nM; p47phox, 134nM) (Figure 2C,D). To examine the interacted domain in MPT64 with TKB1 and HK2, we transfected truncated GST-MPT64 and wild-type Flag-TBK1 or Flag-HK2 in 293T cell and carried out a GST pull-down assay. The N-terminal domain of MPT64 interacted with TBK1, and the core peptide sequences for binding were 24–28 and 34–38 peptides (Figure 2E,F). The C-terminal of MPT64 was associated with HK2 and its essential peptides were located between 187–193 (Figure 2G,H). Additionally, we transfected the wild-type or truncated TBK1 or HK2 with the wild-type Myc-MPT64 to find the interaction site in TBK1 and HK2. The kinase domain (1–301) in TBK1 and HK2 domain in HK2 were found to be essential for their interaction with MPT64 (Appendix A). In summary, these results showed that MPT64 associated with TBK1 and HK2 through the N- or C-terminal in MPT64.

### 3.3. TBK1-Derived Peptide Has Antimycobacterial Activity via the STING1–TBK1–IRF3 Pathway

To examine the role of the interaction between MPT63 or/and MPT64 and TBK1, we constructed wild-type or mutant TBK1-derived peptides, including the interacted domain with MPT63 or/and MPT64 shown in previous results (R9, TBK1 peptide (MPT63, MPT64, and MPT63/64)) (Figure 3A). From our previous results, TBK1 binds with MPT63 or MPT64, so we compared the Kd values of the MPT63 or MPT64 peptides that bind TBK1. Peptides that bind TBK1 have a higher affinity for MPT64 than MPT63 (Appendix A). TBK1 is a vital component of the STING1–TBK1–IRF3 pathway and forms a complex, which induces the expression of IFN-β [34]. To investigate the interaction with the components of the STING1–TBK1–IRF3 pathways, we co-transfected Flag-TBK1, V5-IRF3, and AU1-STING1 with TBK1 peptides. The interactions of TBK1 with IRF3 and STING1 were inhibited by the MPT63/64-TBK1 peptide but the interaction between IRF3 and STING1 was not (Figure 3B). MTB upregulated the activation of STING1–TBK1–IRF3 by increasing the expression of IFN-β for survival in the host, by regulating inflammation [35]. The antimycobacterial activity of MPT63/64-TBK1 was evaluated using MTB-infected THP-1 cells. The complex of STING1–TBK1–IRF3 decreased in MPT63/64-TBK1 in a dose-dependent manner (Figure 3C). In bone marrow-derived macrophages (BMDMs), the knockout of TBK1 showed no interaction with IRF3 and STING1, irrespective of the dose of MPT63/64-TBK1 peptide given (Figure 3D). The increase in the level of pro-inflammatory cytokines, including TNF-α and IL-6, is essential for the bactericidal effect in the host. MTB downregulates the expression of pro-inflammatory cytokines by evading the host immune response [2]. Treatment with MPT63/64-TBK1 decreased the secretion of IFN-β and increased the levels of TNF-α and IL-6 in a dose-dependent manner (Figure 3E). Furthermore, MPT63/64-TBK1 peptides also reduced the burden of MTB in TBK^+/+^ not TBK^−/−^ (Figure 3F). Altogether, these results showed the MPT63/64-TBK1 peptide produced an antimycobacterial effect through decreasing the expression of IFN-β and enhancing the level of pro-inflammatory cytokines.

### 3.4. p47 Peptide Eliminates MTB via Increasing the Level of ROS

p47phox is an essential component for the activation of NADPH oxidase by constructing a complex with p22phox and p67phox. The activation of NADPH oxidase induces the production of cellular ROS. ROS are important for the immune response toward bacterial elimination [36]. We constructed the p47 peptide, including interacted amino acids, with MPT63 to investigate the binding between p47phox and p22phox with treatment of p47phox in 293T cells. The interaction between p47phox and p22phox or p67phox increased with p47 peptide in a dose-dependent manner (Figure 4A). For endogenous binding among p47phox–p22phox–p67phox, we treated p47 peptide and co-immunoprecipitated in THP-1 and BMDMs. The formation of complex of p47phox–p22phox–p67phox was augmented with higher concentration of p47 peptide, although this was not seen in p47phox^−/−^ BMDM cells (Figure 4B). In addition, the stability of p47phox was increased with treatment with p47 peptide. However, the p47 mutant peptide, which replaced the essential amino acids (E152, D153, and E158) to alanine, did not enhance the stability of p47phox (Figure 4C). To investigate the role of the interacted peptide with MPT63 in p47phox in the infection by MTB, we measured the level of cellular or mitochondrial ROS with p47 peptide in MTB-infected BMDM cells. p47 peptide increased the level of cellular ROS but not of mitochondrial ROS in a dose-dependent manner (Figure 4D). Furthermore, the p47 peptide significantly upregulated the level of TNF-α and IL-6 in MTB-infected macrophages, although these results were not observed with the p47 mutant peptide (Figure 4E). The amount of intracellular MTB decreased linearly with higher concentrations of p47 peptide (Figure 4F). With NAC or DPI, the induction of pro-inflammatory cytokines and antimycobacterial effect was decreased in MTB-infected BMDM with p47 peptides (Appendix A). In summary, these results indicated that p47 peptide increases the level of ROS and pro-inflammatory cytokines toward the elimination of MTB via upregulation of the interaction among p47phox–p22phox–p67phox.

### 3.5. HK2 Peptide Specifically Targets the MTB-Infected Macrophages

HK2 plays an essential role in glycolysis metabolism, converting D-glucose to α-D-glucose-6-phosphate. The glycolysis metabolism is intimately linked to inflammation, and this is known as immunometabolism. Previous studies showed that HK2 accumulates within an inflammatory environment, to activate an immune response [33,37]. We examined the binding of HK2 peptide with HK2 in MTB-infected macrophages. We constructed the HK2 peptide, including the domain of MPT64 that interacted with HK2. To investigate the interaction between HK2 peptide and HK2, the number of Cy5.5 labelled-HK2 peptide^+^ cells was counted using flow cytometry. Interestingly, HK2 peptide only interacted with HK2 and not HK1 and HK3 in macrophages (Figure 5A). To examine the effect of HK2 peptide in MTB infection, we treated the HK2 peptide in MTB-infected macrophage. The secretion of pro-inflammatory cytokines was not increased and the burden of bacteria did not decrease significantly (Figure 5B). To evaluate the specificity of HK2 peptide, we administered the HK2-peptide via intranasal injection in MTB-infected mice. HK2 targets macrophages in MTB-infected lungs but not in other immune cells (Figure 5C). In summary, these results showed that HK2 binds to MTB-infected macrophages.

### 3.6. Recombinant Multifunctional MPT Protein Increases the Mycobactericidal Effect in Macrophages

Previous results showed that TBK1 and p47 peptides increased the expression of pro-inflammatory cytokines and the mycobactericidal activity in macrophages. Additionally, HK2 peptide increased the specific interaction with HK2 of macrophages in MTB infection. To generate the protein that included the functions of TBK1, p47, and HK2 peptide, we designed a recombinant multifunctional MPT protein (rMPT), containing the poly-sequences of TBK1, p47, and HK2 peptide. This was confirmed using SDS-polyacrylamide gel electrophoresis and immunoblotting (Figure 6A,B). No significant differences, compared with the vector controls, were observed regarding rMPT-induced cytotoxicity in BMDMs (Figure 6C). To examine the localization of MPT and its binding partners, we treated rVehicle or rMPT in BMDMs and results were observed by fluorescent images. Contrarily to rVehicle, rMPT co-localized with HK2, p47phox, and TBK1 in BMDMS (Figure 6D). Furthermore, rMPT interacted with HK2, p47phox, and TBK1 in a dose-dependent manner. Phosphorylated forms of p47phox (S345 and S359) or TBK1 (S172) were also specifically associated with rMPT (Figure 6E and Appendix A). In MTB-infected macrophages, the complex STING1–TBK1–IRF3 was consistently decreased by rMPT, consistent with Figure 3C,D (Figure 6F), and the assembly of p47phox–p22phox–p67phox was increased following the treatment of rMPT in MTB-infected BMDMs (Figure 6G). The secretion of IFN-β was downregulated, contrarily to TNF-α and IL-6, by rMPT in Wild type (WT). In BMDM of TBK1^−/−^ and p47phox^−/−^ or Cre-induced HK2 KO mice, there were no significant differences between the control and treatment rMPT in MTB infection (Figure 6H and Appendix A). The bacterial burden declined in rMPT-treated WT macrophages, but not in TBK1^−/−^, p47phox^−/−^, or Cre-induced HK2 KO BMDMs (Figure 6I). Additionally, we constructed rMPT without HK2 peptide to examine whether the HK2 peptide is essential in rMPT for specifically targeting the macrophages (Appendix A). In MTB-infected mice, rMPT specifically targeted the macrophages and DCs but not rMPT without HK2 peptide (Appendix A). These results demonstrated that rMPT interacts with HK2, p47phox, and TBK1, and reduces the burden of MTB by activating the inflammatory response. 

### 3.7. rMPT Increases the Vaccination against MTB Infection in Mice

The BCG vaccine is important for preventing TB, but the effect of BCG vaccine is lost in adult pulmonary TB. Therefore, the development of a potential vaccine candidate is urgent [38]. To identify the vaccinating ability of rMPT in TB, we treated rMPT with DDA-MPL (adjuvant) in BCG-infected mice. After vaccination, we infected MTB through an intranasal injection to the mice (Figure 7A). In the lung and spleen, the bacterial burden was decreased, except in the non-treated group in WT mice. Surprisingly, mice vaccinated with BCG and rMPT showed a more significant reduction of burden of MTB, when compared to other groups. The deficiency of TBK1, p47phox, and HK2 in mice showed no effect on the vaccination of rMPT (Figure 7B). To examine the rMPT-induced reactivation of adaptive immunity, we restimulated the lung cells ex vivo with purified protein derivative (PPD) or rMPT. BCG or BCG+rMPT-vaccinated lung cells were activated by PPD, although this was not observed in rMPT-vaccinated lung cells. Contrarily to BCG-vaccinated lung cells, rMPT-vaccinated lung cells were activated in rMPT re-stimulation (Figure 7C). This indicated that rMPT increases the vaccination and reactivation of the adaptive immunity against TB.

### 3.8. rMPT Downregulates the Survival of MTB in Mice

We aimed to determine whether rMPT increases the antimycobacterial effect in mice. We infected mice with MTB H37Rv strain followed with rMPT treatment through an intranasal injection (Figure 8A). First, we confirmed the number of intracellular bacteria and evaluated the pathology in the lungs. The bacterial count decreased and the infiltration of immune cells and damage to the lungs was reduced in rMPT-treated mice (Figure 8B and Appendix A). The bacterial colony forming units (CFUs) and granulomas were reduced in rMPT-treated mice; however this difference was not significant in mice lacking TBK1, p47phox, or HK2, compared to the control (Figure 8C). Consistently with Figure 6E, rMPT was associated with TBK1, p47phox, and HK2 in the lungs of MTB-infected mice (Figure 8D). Furthermore, we measured the biodistribution and pharmacokinetics of rMPT in mice using an in vivo imaging system (IVIS) spectrum-chromatography (CT) system. rMPT was accumulated in the lung and spleen within 1 h. Additionally, rMPT was excreted in the liver within 6 h, whereas rMPT remained in the lung over 24 h (Figure 8E). In summary, rMPT increases the antimycobacterial effects through interacting with TBK1, p47phox, and HK2 in MTB-infected mice.

## 4. Discussion

The present study describes a novel antimycobacterial candidate for use as a vaccine and therapeutic based on MPT63 and MPT64, via regulating the expression of IFN-β and production of ROS and specifically targeting the infected macrophages. The major findings of this study are as follows: (1) MPT63 interacted with TBK1 and p47phox, and MPT64 was immediately associated with TBK1 and HK2; (2) TBK1 peptide downregulated the production of IFN-β and the burden of bacteria, through the inhibition of the STING1–TBK1–IRF3 pathway in MTB-infected macrophages; (3) p47 peptide increased the levels of ROS and eliminated MTB in macrophages, through enhancing the construction of the complex of p47phox–p22phox–p67phox; (4) HK2 peptide was able to target MTB-infected macrophage specifically; (5) rMPT, constructed with the functions of TBK1, p47, and HK2 peptides, showed antimycobacterial activity by regulating the level of IFN-β and ROS in MTB-infected macrophages; (6) rMPT elevated the effects of the vaccine and therapy against MTB infection in vivo (Figure 9).

The secreted effector proteins from MTB are essential for the virulence in the host, by interacting with target factors of the host and regulating the signaling pathway. Numerous secretory effectors are secreted by five type-seven secretion systems, ESX-1 to ESX-5. Secretory antigens of MTB are essential for understanding the pathogenesis of TB [4]. The early secreted antigenic target of 6 kDa (ESAT-6), a well-known antigen, is one of the secretory proteins secreted through the ESX-1 system [39]. ESAT-6 forms a heterodimer with culture filtrate proteins of 10 kDa (CFP10) encoding the same genome locus, called the RD1 region [40]. Secreted ESAT-6 is involved in the virulence of MTB via regulating the host proteins. There is evidence that ESAT-6 disrupts the phagosome maturation by inhibiting the fusion of lysosome with phagosome [41]. Likewise, the secretion of MPT63 and MPT64 is increased in active TB. The ortholog MPB63 from *M. bovis* only differs in nucleotide sequence from MTB MPT63 but is not distinct in function [42]. MPT64, the gene locus in RD2, is not expressed in the BCG strain because of the lack of RD2 region [43]. In previous studies, these proteins, among other secreted antigens and candidates for vaccines, were shown to be promising serological markers [12,13,14,15]. MPT63 plays a crucial role in determining the pathway of antigen presentation and recognition by Th1 cell. MPT63 and its epitope bind promiscuously to HLA-DR, an MHC class II cell surface receptor, and induce a moderate Th1 cell reactivity [44]. MPT64 is also associated with the expression of IFN-γ in activating murine macrophages. Furthermore, MPT64 is related to the expression of apoptotic factors, such as Caspase 3, FasL, Fas, and Bax in epithelioid cells and multinucleated giant cells [45]. In our study, we showed that MPT63 associated with the host immune response with TBK1 and p47phox regulating the level of IFN-β and ROS in TB infection. Moreover, MPT64 regulated the expression of IFN-β via interaction with TBK1. These results increase the understanding of the importance of the interaction between secretory antigens of MTB and host proteins.

IFN-β is a crucial cytokine for protection from many pathogens, including viruses, bacteria, and protozoa, and plays an essential role in the induction of the innate and adaptive immune response [46,47,48,49]. In contrast to the protective role of IFN-β in anti-viral immune response, MTB utilizes IFN-β to increase its survival in the host [50,51]. Recent studies have reported that IFN-β increases pro-bacterial activity, correlated with enhanced anti-inflammatory properties [52]. IFN-β is an antagonist of IL-1β and IL-18, as it increases the expression of IL-10 and disrupts the assembly of NLRP3 inflammasome [21]. MTB secretes its chromosomal DNA to induce an IFN-β response by being recognized by cyclic GMP-AMP synthase (cGAS, a DNA sensor). Studies have described that cGAS is required for the activation of the expression of IFN-β via the STING1-TBK1-IRF3 pathway during MTB infection [53,54]. In the present research, we reported that rMPT regulates the survival of MTB through alleviating the production of IFN-β.

ROS are powerful microbicidal factors in immune responses. In cytosol, ROS are produced by NADPH oxidase (NOX2), which consists of several subunits, including gp91phox, p22phox, p47phox, p67phox, and Rac1. gp91phox and gp22phox are localized on the phagosome membrane; p47phox, p67phox, and Rac1 are recruited to gp91 and the active complex generates ROS through a redox chain [55]. In MTB infection, NADPH oxidase is activated to burst the production of ROS to control the intracellular bacterial load through its bactericidal activity and induces apoptosis [56,57]. For survival in the host, MTB inhibits NOX2 activity in different ways. MTB nucleoside diphosphate kinase (Ndk) acts as GTPase-activating protein that inhibits the NOX2-related GTPase and Rac1, which is essential for the assembly of NOX2. Knockdown of Ndk decreases the survival of MTB in macrophages [58,59]. KatG is a catalase produced by MTB. KatG, in a previous study, was shown to counter the oxidative burst in phagocytes during mycobacterial pathogenesis. The mutation of KatG MTB strain was eliminated in murine BMDMs [23]. 

Hexokinases (HKs) phosphorylate glucose to glucose 6-phosphate in glucose metabolism. HK2, one of the isoforms of hexokinases, is not only essential as a glycolytic enzyme but is also a mediator of autophagy. HK2 controls mTORC1, which is the main regulator of cell growth and autophagy, dependent on conditions of nutrition [60]. Furthermore, HK2 is a pivotal factor in cancer metabolism, called the “Warburg effect”; the active tumor increases the expression of HK2 by upregulating the glycolysis metabolism for survival [61]. In TB, the glycolysis metabolism is elevated through the expression of glycolytic enzymes, including HK2. Normally, this increase is linked to the activation of inflammation toward the host defense mechanisms against bacteria. Indeed, MTB exploits the host metabolism through glycolysis and lipid metabolism for survival and proliferation in the host. However, the role of HK2 in TB remains unclear [33,37,62].

Furthermore, this study showed that the kinase domain of TBK1, PX domain in p47phox, and HK2 domain in HK2 are essential for interacting with MPT63 and/or MPT64. The kinase domain in TBK1 is essential for activation of TBK1 to phosphorylate the downstream factors [63,64,65]. Some amino acid residues related to kinase activity include an active site as proton acceptor [63,65]. This study showed TBK1 peptide domains in rMPT inhibited the role of TBK1. We assume that these domains may interact with these residues with their essential amino acids for binding with TBK1. PX domain is known to bind to phosphatidylinositol 3,4-bisphosphate and anionic phospholipids to enhance membrane affinity. Membrane bound-p47phox formed an open conformation by exposing the SH3 domains that facilitate the p22phox to construct the activated NADPH oxidase [66,67]. We speculated that the essential peptide of MPT63 may help to form the open conformation of p47phox but further study is needed to evaluate how it plays a role in p47phox activation. HK2 domain is important for catalytic activity and binding with substrate in HK2. Although HK2 was considered as a target against cancer, it is ambiguous how HK2 is targeted in MTB infection [68,69]. Interestingly, HK2 peptide in rMPT specifically targeted the macrophages and DCs in TB infection in this study. Further study of the mechanisms of HK2 peptide in rMPT is needed for the development of specific therapeutic agents against TB.

Here, we studied the role in MPT63 and MPT64 to detect putative binding and examined the mechanism of regulation of expression of IFN-β and production of ROS through the use of TBK1 and p47 peptide in MTB-infected macrophages. Although HK2 peptide showed no significant effect in controlling the MTB load, it specifically targeted the macrophages via binding to HK2. We speculated that the HK2 peptide is similar to signal peptide for targeting macrophages in MTB infection. The multifunctional rMPT controlled the survival of MTB via upregulating the inflammation in vitro and in vivo. It is important to note that rMPT may be a potential candidate both as a vaccine and a therapeutic agent against MTB. Additionally, the construction of a multifunctional protein derived from pathogens has previously allowed the development of novel therapeutic agents in diverse diseases. We previously studied the role of *Toxoplasma gondii* dense granule proteins in host macrophages and other studies were able to develop candidates for therapeutic agents against other diseases, including TB, sepsis, and cancer [27,70,71,72]. The present study suggests that MPT63 and MPT64 mediate the regulation of IFN-β and ROS in the host macrophages and that rMPT derived from MPT63 and MPT64 may be a candidate for a novel vaccine or a therapeutic agent against TB.

## Figures and Tables

**Figure 1 biomedicines-09-00545-f001:**
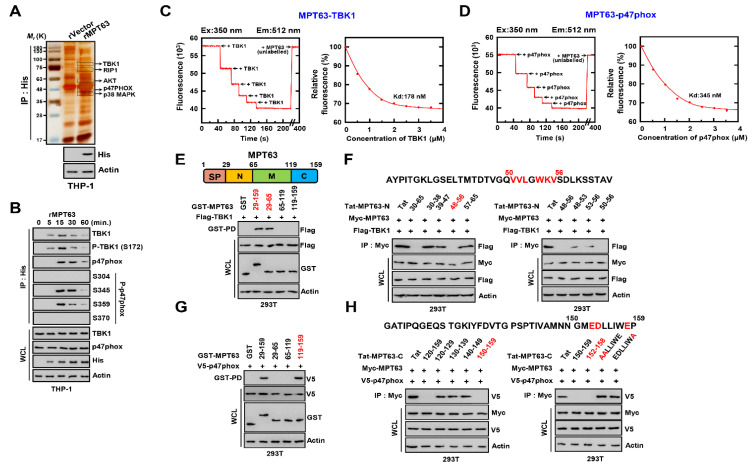
TBK1 and p47phox is bound with MPT63. (**A**) Identification of TBK1 and p47phox by mass spectrometry analysis in THP-1 cell lysates treated with rMPT63 or rVector. (**B**) THP-1 cells were stimulated with rMPT63 (5 μg mL^−1^) for the indicated times, followed by immunoprecipitation (IP) with αHis-agarose bead and IB with αTBK1, αP-TBK1 (S172), αp47phox, αP-p47phox (S304, 345, 359, and 370), αHis, αActin. (**C**,**D**) Titration of fluorescently labelled MPT63 with TBK1 and p47phox (left), with K_d_ (178 and 345 nM) determined by curve fit analysis (right). (**E**,**G**) Binding mapping. Schematic diagrams of the structures of MPT63 (upper). At 48 h after transfection with mammalian glutathione S-transferase (GST) or GST-MPT63 and truncated mutant constructs together with Flag-TBK1 or V5-p47phox. 293T cells were used for GST pull down, followed by IB with αFlag or αV5. Cell lysates (WCLs) were used for IB with αFlag or αV5, αGST, and αActin. (**F**,**H**) 293T cells are expressing Myc-MPT63 and Flag-TBK1 or V5-p47phox and treated with several Tat-MPT63-N or MPT63-C peptides (10 µM) for 6 h, followed by IP with αMyc and IB with αFlag. WCLs were used for IB with αMyc, αFlag, and αActin. The data are representative of four independent experiments with similar results (**A**–**H**).

**Figure 2 biomedicines-09-00545-f002:**
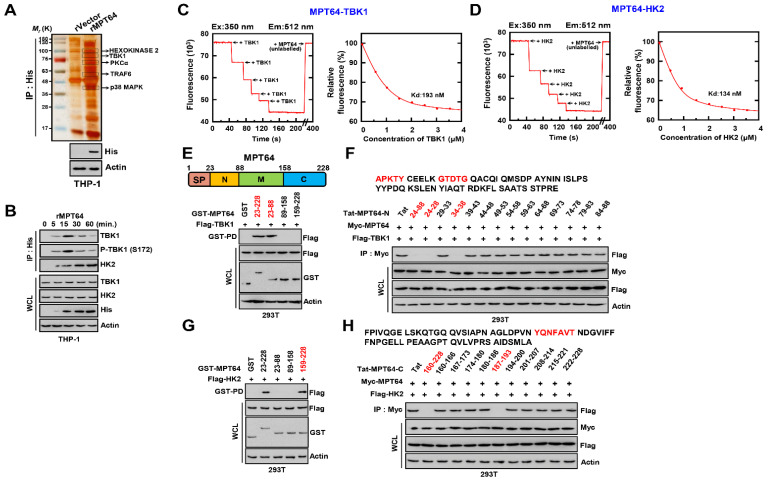
TBK1 and HK2 directly interact with MPT64. (**A**) Identification of TBK1 and HK2 by mass spectrometry analysis in THP-1 cell lysates treated with rMPT64 or rVector. (**B**) THP-1 cells were stimulated with rMPT64 (5 μg mL^−1^) for the indicated times, followed by IP with αHis-agarose bead and IB with αTBK1, αP-TBK1 (S172), αHK2, αHis, αActin. (**C**,**D**) Titration of fluorescently labelled MPT64 with TBK1 and HK2 (left), with K_d_ (193 and 134 nM), determined by curve fit analysis (right). (**E**,**G**) Binding mapping. Schematic diagrams of the structures of MPT64 (upper). At 48 h after transfection with GST or GST-MPT63 and truncated mutant constructs together with Flag-TBK1 or V5-p47phox. 293T cells were used for GST pull down, followed by IB with αFlag or αV5. WCLs were used for IB with αFlag or αV5, αGST, and αActin. (**F**,**H**) 293T cells expressing Myc-MPT63 and Flag-TBK1 or V5-p47phox and treated with several Tat-MPT64-N or MPT64-C peptides (10 µM) for 6 h, followed by IP with αMyc and IB with αFlag. WCLs were used for IB with αMyc, αFlag, and αActin. The data are representative of four independent experiments with similar results (**A**–**H**).

**Figure 3 biomedicines-09-00545-f003:**
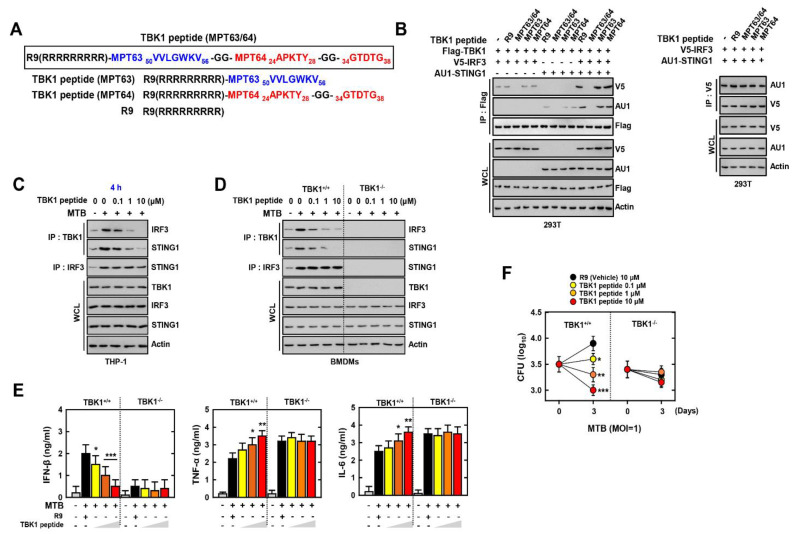
TBK1 peptide eliminates MTB by decreasing the STING1–TBK1–IRF3 pathway. (**A**) Schematic design of TBK1 peptide and its mutants. (**B**) 293T cells were transfected with V5-IRF3 and/or AU1-STING1 and treated TBK1 peptide or its mutants for 6h (1 µM) with Flag-TBK1 (left) or without Flag-TBK1 (right). 293T cells were used for IP with αFlag or αV5, followed by IB with αV5, αAU1 and αFlag. WCLs were used for IB with αV5, αAU1, αFlag, and αActin. (**C**,**D**) THP-1, TBK1^+/+^ or TBK1^−/−^ BMDM cells were infected by MTB for 4 h and treated TBK1 peptide at various concentrations. After 4 h, THP-1 cells were used for IP with αTBK1 or αIRF3, followed by IB with αIRF3 and αSTING1. WCLs were used for IB with αTBK1, αIRF3, αSTING1, and αActin. (**E**) TBK1^+/+^ or TBK1^−/−^ BMDM cells were infected by MTB for 4 h and treated TBK1 peptide at various concentrations. After 18h, supernatants of BMDMs were used for analysis of the level of IFN-β, TNF-α, and IL-6 by ELISA. (**F**) The burden of MTB was evaluated after 3 d in MTB infected TBK1^+/+^ or TBK1^−/−^ BMDMs with vehicle or TBK1 peptide. The data are representative of four independent experiments with similar results (**B**–**F**). Significant differences (* *p* < 0.05; ** *p* < 0.01; *** *p* < 0.001) compared with rVehicle-treated BMDMs.

**Figure 4 biomedicines-09-00545-f004:**
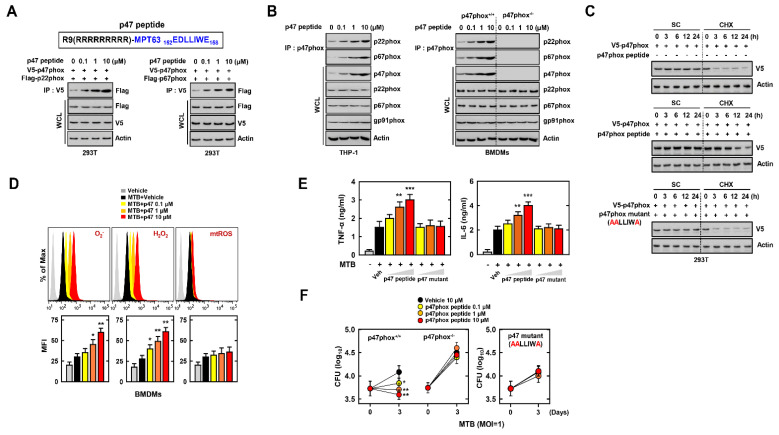
p47 peptide enhances inflammation against MTB by elevating the level of ROS. (**A**) Schematic design of p47 peptide (upper). 293T cells were transfected with V5-p47phox and Flag-p22phox or Flag-p67phox and treated p47 peptide for 6 h (1 µM). 293T cells were used for IP with αV5, followed by IB with αFlag. WCLs were used for IB with αFlag, αV5, and αActin. (**B**) THP-1 or BMDM cells were treated with p47phox in the indicated concentrations. After 18 h, THP-1 or BMDMs cells were used for IP with αp47phox, followed by IB with αp22phox and αp67phox. WCLs were used for IB with αp47phox, αp22phox, αp67phox, αgp91phox, and αActin. (**C**) p47 peptide mediates the increase of ATP5A1 stability. 293T cells were transfected with V5-p47phox and treated p47 peptide for 24 h. After 24 h, 293T cells were treated with solvent control (SC) or cyclohexamide (CHX, 1 μg mL^−1^) for the indicated times and cell lysates were used for IB with αV5 and αActin. (**D**) BMDMs were infected MTB for 4 h and treated Vehicle or p47 peptide of various concentrations for 18 h. To examine the level of ROS, BMDMs measured the fluorescence of DHE, DCFH-DA, mitoSOX to detect O_2_^−^, H_2_O_2,_ and mtROS by FACS. (**E**) BMDMs were infected with MTB for 4 h and treated Vehicle, p47 peptide, or p47 mutant of various concentrations. The supernatant of BMDMs were used for ELISA to measure the level of TNF-α and IL-6. (**F**) The burdens of MTB in p47 peptide treated-BMDMs were measured after 3 d. The data are representative of four independent experiments with similar results (**B**–**F**). Significant differences (* *p* < 0.05; ** *p* < 0.01; *** *p* < 0.001) compared with vehicle-treated BMDMs.

**Figure 5 biomedicines-09-00545-f005:**
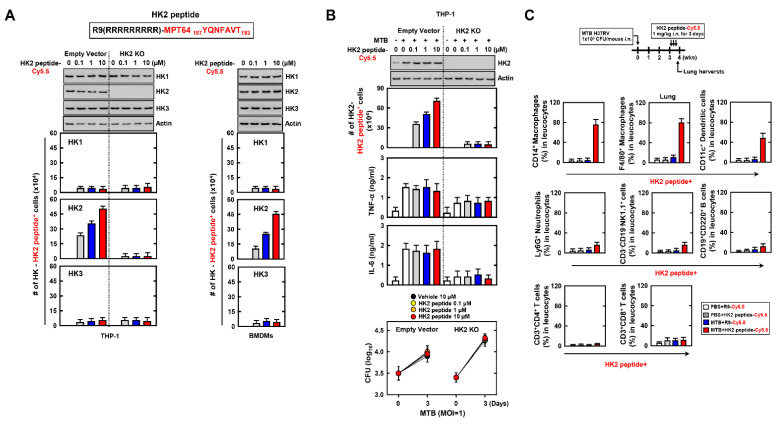
HK2 peptide’s role in signal peptide for targeting the MTB-infected macrophages. (**A**) Schematic design of HK2 peptide (upper). Empty or HK2 KO THP-1 and BMDM cells were treated with Cy5.5 labelled-HK2 peptide for 1 h in various concentrations. THP-1 and BMDMs were used for IB or counting the number of HK-HK2 peptide^+^ cells by FACS. (**B**) Empty or HK2 KO THP-1 cells were infected with MTB for 4 h and treated Cy5.5 labelled-HK2 peptide in various concentrations for 1, 18 or 72 h. After 1 h, the THP-1 cells were used for IB and the number of HK2-Hk2 peptide^+^ cells were counted by FACS (top). The HK2 peptide treated-supernatants of THP-1 for 18 h were used for ELISA to measure the level of TNF-α and IL-6 (middle). The colony forming units (CFU) of intracellular MTB in THP-1 cells were measured after 3 d (bottom). (**C**) Mice was infected by MTB through intranasal infection (1 × 10^3^/per mice) and intranasally treated Cy5.5-labelled HK2 peptide (1 mg kg^−1^) after 3 wks. Lung harvests were used for analysis of the number of HK2 peptide^+^ cells by FACS. The data are representative of four independent experiments with similar results (**A**–**C**).

**Figure 6 biomedicines-09-00545-f006:**
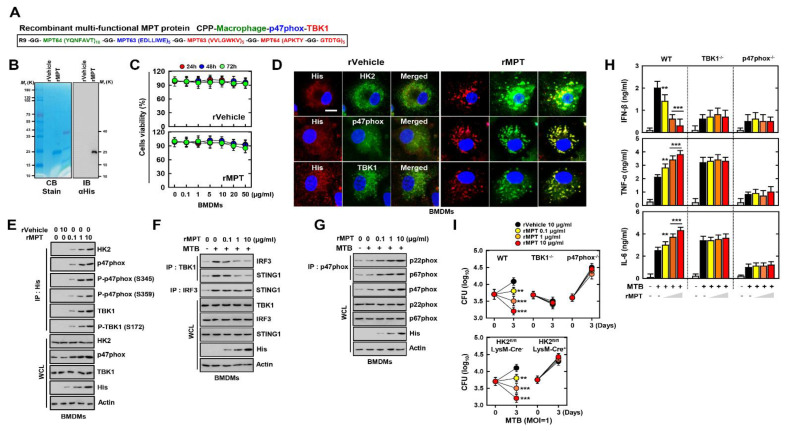
rMPT regulates MTB infection through enhancing the inflammation with declining the expression of IFN-β and increasing the level of ROS in macrophages. (**A**) Schematic in design of rMPT. (**B**) Bacterially purified 6xHis-rMPT and rVehicle were analyzed by coomassie blue staining (left) or immunoblotting (IB) with αHis (right). (**C**) BMDMs were incubated with rVehicle and rMPT for the indicated times and concentrations, then cell viability was measured with MTT assay. (**D**) BMDMs were treated with rVehicle or rMPT and immunolabelled with αHis (Alexa 586), αHK2, α p47phox, αTBK1 (Alexa 488), and DAPI. Scale bar, 10 μm. (**E**) BMDMs were treated with rVehicle or rMPT for 1 h. BMDMs were used for IP by αHis, followed by IB with αHK2, αp47phox, αP-p47phox (S345 and S359), αTBK1, αP-TBK1 (S172). WCLs were used for IB with αHK2, αp47phox, αTBK1, αHis, and αActin. (**F**) BMDMs were infected by MTB for 4 h and treated rMPT in various concentrations for 1 h. BMDMs were used for IP by αTBK1 and αIRF3, followed by IB with αIRF3 and αSTING1. WCLs were used for IB with αTBK1, αIRF3, αSTING1, αHis, and αActin. (**G**) BMDMs were used for IP by αp47phox, followed by IB with αp22phox and αp67phox. WCLs were used for IB with αp47phox, αp22phox, αp67phox, αHis, and αActin. (**H**) WT, TBK^−^^/^^−^, or p47phox^−^^/^^−^ BMDMs were infected by MTB for 4 h and treated rMPT in various concentrations for 18 h. The supernatant of BMDMs were used for ELISA to measure the level of IFN-β, TNF-α, and IL-6. (**I**) The load of intracellular bacteria was measured after 3 d from treating the rVehicle or rMPT in WT, TBK^−^^/^^−^ or p47phox ^−^^/^^−^ (Upper) and HK2^fl/fl^ LysM-Cre^‑^, or HK2^fl/fl^ LysM-Cre^+^ BMDMs. The data are representative of four independent experiments with similar results (**C**–**I**). Significant differences (* *p* < 0.05; ** *p* < 0.01; *** *p* < 0.001) compared with rVehicle-treated BMDMs.

**Figure 7 biomedicines-09-00545-f007:**
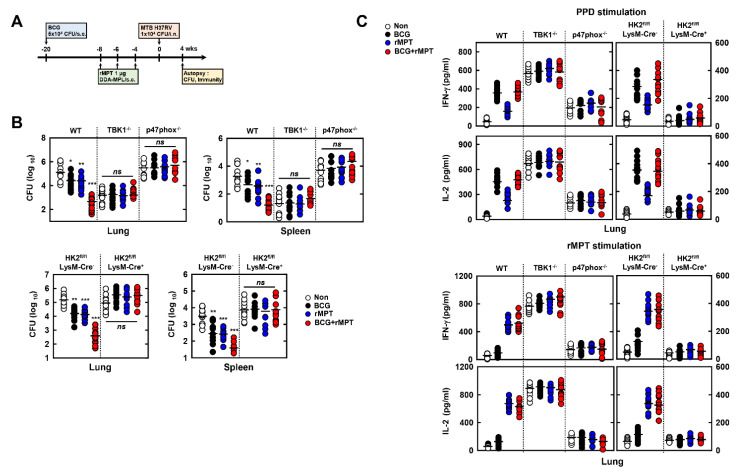
rMPT increases the vaccination against MTB in mice. (**A**) Schematic of the vaccine testing model treated with rMPT in mice. Mice (*n* = 10 per group) were immunized by BCG through subcutaneous injection 12 wks before vaccinating with rMPT (1 μg). Three subcutaneous injection of rMPT with DDA-MPL (adjuvant) were administered before MTB H37Rv intranasal infection. Immunological analysis was carried out after 4 wks. (**B**) CFU in the lungs and spleen in all groups at 4 wks post-infection. (**C**) Mice of each group were sacrificed 4 weeks post-infection, followed by obtaining the lung harvests and stimulating with purified proteins derivative (PPD, 10 μg/mL) or rMPT (0.1 μg/mL) in each group. The supernatants were used for measuring the level of IFN-γ and IL-2 by ELISA. The data are representative of four independent experiments with similar results (**B**,**C**). Significant differences (* *p* < 0.05, ** *p* < 0.01; *** *p* < 0.001) compared with non-treated mice.

**Figure 8 biomedicines-09-00545-f008:**
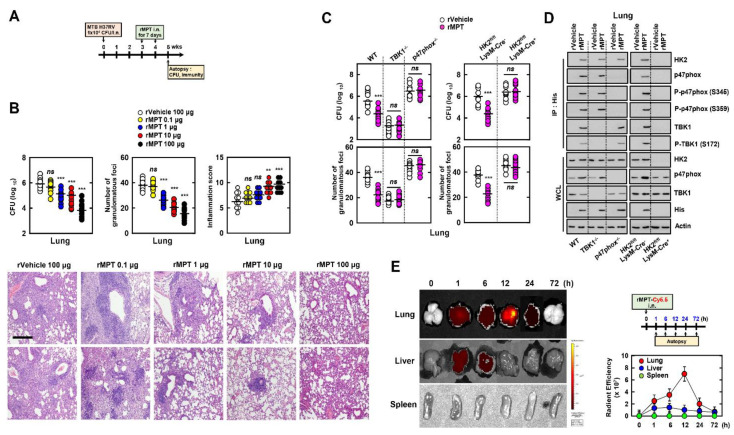
rMPT is a potential therapeutic agent against MTB in mice. (**A**) Schematic of TB model treated with rMPT or rVehicle. Mice (*n* = 10 per group) were intranasally infected by MTB H37Rv (1 × 10^4^ CFU/mice). After 3 wks, mice were treated with rMPT or rVehicle for 7 d. Immunological analysis conducted in 5 wks. (**B**) Bacterial loads, the number of granuloma, and the level of inflammation were analyzed in each group of mice lungs (upper). Histopathology scores were obtained from H&E stained lung sections (bottom). Scale bar, 500 μm. (**C**) Bacterial loads were counted in WT, TBK^−/^^−^, and p47phox ^−/^^−^ HK2^fl/fl^ LysM-Cre^−^, and HK2^fl/fl^ LysM-Cre^+^ mice lung. (**D**) Lung harvests in each group of mice were used for IP with His-agarose bead, followed by IB with αHK2, αp47phox, αP-p47phox (S345 and S359), αTBK1, and αP-TBK1 (S172). WCLs were used for IB with αHK2, αp47phox, αTBK1, αHis, and αActin. (**E**) Fluorescence images of the lung, liver, and spleen of the mice intranasally administrated with Cy5.5 labelled-rMPT (left), and quantitative fluorescence intensities of the organs measured by an IVIS spectrum-chromatography (CT) system. The data are representative of four independent experiments with similar results (**B**–**E**). Significant differences (* *p* < 0.05, ** *p* < 0.01; *** *p* < 0.001) compared with rVector-treated mice.

**Figure 9 biomedicines-09-00545-f009:**
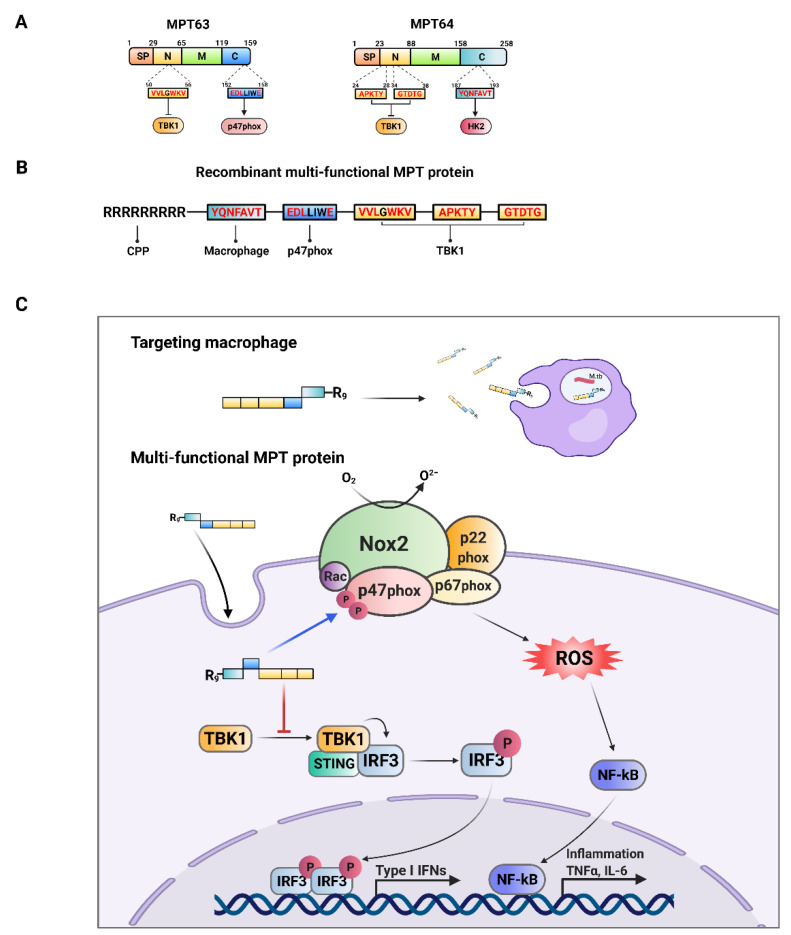
Schematic model for the roles in rMPT against MTB infection. (**A**) Domain screening of interacting site between MPT63 or MPT64 with TBK1, p47phox and HK2. (**B**) Construction of rMPT combined TBK1, p47phox, and HK2 interacting domains in MPT63 and MPT64. (**C**) Regulatory pathway of rMPT in macrophages.

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
