# Peer review of "Multi-Functional MPT Protein as a Therapeutic Agent against Mycobacterium tuberculosis"

_biomedicines, 2021, doi:10.3390/biomedicines9050545_

Round 1

Reviewer 1 Report

The manuscript by Kim et al. entitled “Multi-functional MPT protein as a therapeutic agent against Mycobacterium tuberculosis” seeks to identify M. tb’s secretion proteins MPT63 and MPT64 as a set of virulence factors, with which M. tb regulates the host antimicrobial signaling activities. The authors revealed MPT mediated advantageous survival and replication of M. tb within hosts is attributed to the regulatory interaction with host TBK1, p47phox, and HK2. Based upon these findings, authors constructed rMPT peptide (rMPT) to interfere with functions of MPT63 and MPT64 and to propose the peptide as a vaccine or a therapeutic candidate for tuberculosis infection.

The authors present a variety of experiments to demonstrate convincingly that M.tb MPT family is a virulent factor to regulate host antimicrobial signaling for the intracellular survival and replication. Despite that, there are some issues for further consideration.

1- TBK1 serves as a common binding partner of both MPT63 and MPT64. The Kd values of TBK1 against MPT63 and MPT64 were very similar. Thus, it would be interesting to identify which MPT protein has a greater binding activity to TBK1 to activate STING-TBK1-IRF3 signaling and IFN-beta production. This can be assessed by TBK1-MPT63 peptide and TBK1-MPT64 peptide.

Line 169 – To examine the role of the interaction between MPT64 and TBK1: why only MPT64 not MPT63 ?

2- MPT63 and MPT64 structures of M. tb were well defined. Now, authors identified the specific amino acids that are involved in binding TBK1 and p47phox or HK2. It would be interesting if authors explain the functional relevance of the amino acids in the structural perspectives in discussion section.

3- There are many syntax errors. Please revise them carefully.

4- 2.2 subtitle contains “directly”. Does it mean the interaction between MPT63 and TBK1 or p47phox is not a direct interaction ?

5- Fig. 3B: IP:Flag Flag tag IB raw lane number is unmatched.

6- The effect of p47 peptide on proinflammatory cytokine production and antimicrobial activity should be confirmed by treatment with ROS scavenger such as NAC.

7- The reason why rMPT contained HK2 peptide is to make it enrich at the Mtb infected macrophages. If authors include the result of rMPT without HK2 peptide, it may support the rMPT construction scheme why rMPT requires HK2 peptide.

8- In discussion section, authors may compare the phenotype of intracellular Mtb in the presence of rMPT peptide with that of ESX secretion system deficient Mtb. This may help define the relative contribution of MPT mediated evasion of antimicrobial activity to the perturbed viability or attenuated virulence of ESX secretion system deficient Mtb.

9- Line 175 - 176: the effect of MPT63/64 TBK1 peptide on interaction between IRF3 and STING1 is validated in Fig 3C not Fig. 3B.

10- Line 331: Text says Mtb is infected intranasally. Figure legend says Mtb is infected via aerosol infection.

11- Line 394: rMPT; Line 431: Our – our

Author Response

Þ Thanks for your kind and excellent comments. We submit a revised version of our manuscript and a point-by-point response to the reviewers’ comments. Detailed responses are described below.

Reviewer 1
The manuscript by Kim et al. entitled “Multi-functional MPT protein as a therapeutic agent against Mycobacterium tuberculosis” seeks to identify M. tb’s secretion proteins MPT63 and MPT64 as a set of virulence factors, with which M. tb regulates the host antimicrobial signaling activities. The authors revealed MPT mediated advantageous survival and replication of M. tb within hosts is attributed to the regulatory interaction with host TBK1, p47phox, and HK2. Based upon these findings, authors constructed rMPT peptide (rMPT) to interfere with functions of MPT63 and MPT64 and to propose the peptide as a vaccine or a therapeutic candidate for tuberculosis infection.

The authors present a variety of experiments to demonstrate convincingly that M.tb MPT family is a virulent factor to regulate host antimicrobial signaling for the intracellular survival and replication. Despite that, there are some issues for further consideration.

1- TBK1 serves as a common binding partner of both MPT63 and MPT64. The Kd values of TBK1 against MPT63 and MPT64 were very similar. Thus, it would be interesting to identify which MPT protein has a greater binding activity to TBK1 to activate STING-TBK1-IRF3 signaling and IFN-beta production. This can be assessed by TBK1-MPT63 peptide and TBK1-MPT64 peptide.

Line 169 – To examine the role of the interaction between MPT64 and TBK1: why only MPT64 not MPT63?

Þ Thanks for your kind and insightful comments. We have added the supplementary data for TBK1-MPT63 peptide and TBK1-MPT64 peptide for comparing the Kd value (Figure S3). Peptides that bind TBK1 have a higher affinity for MPT64 (Kd : 238 nM) than MPT63 (Kd : 384 nM). We also have revised line 163.

2- MPT63 and MPT64 structures of M. tb were well defined. Now, authors identified the specific amino acids that are involved in binding TBK1 and p47phox or HK2. It would be interesting if authors explain the functional relevance of the amino acids in the structural perspectives in discussion section.

Þ Thanks for your kind and insightful comments. We have added the explanation of binding of MPT63 and MPT64 with TBK1 and p47phox or HK2 in the structural perspectives related the specific amino acids in discussion section.

3- There are many syntax errors. Please revise them carefully.

Þ We are sorry for our mistakes. We have progressed additional english editing by Enago.

4- 2.2 subtitle contains “directly”. Does it mean the interaction between MPT63 and TBK1 or p47phox is not a direct interaction?

Þ We are sorry for our mistakes. We have corrected the title.

5- Fig. 3B: IP:Flag Flag tag IB raw lane number is unmatched.

Þ We are sorry for our mistakes. We have corrected the Fig. 3B and S8 (for whole gel image) data.

6- The effect of p47 peptide on proinflammatory cytokine production and antimicrobial activity should be confirmed by treatment with ROS scavenger such as NAC.

Þ Thanks for your kind and insightful comments. We have supplemented the additional data with ROS scavengers (Figure S4). With NAC or DPI, the induction of pro-inflammatory cytokines and antimycobacterial effect is decreased in MTB-infected BMDM with p47 peptides.

7- The reason why rMPT contained HK2 peptide is to make it enrich at the Mtb infected macrophages. If authors include the result of rMPT without HK2 peptide, it may support the rMPT construction scheme why rMPT requires HK2 peptide.

Þ Thanks for your kind and insightful comments. We have compensated the additional data for examine the significance of HK2 peptide in rMPT against Mtb infection (Figure S6). We constructed rMPT without HK2 peptide to examine whether HK2 peptide is essential in rMPT for specific targeting the macrophages (Figures S6A-C). In MTB-infected mice, rMPT specifically target the macrophages and DCs, but not rMPT without HK2 peptide (Figure S6D).

8- In discussion section, authors may compare the phenotype of intracellular Mtb in the presence of rMPT peptide with that of ESX secretion system deficient Mtb. This may help define the relative contribution of MPT mediated evasion of antimicrobial activity to the perturbed viability or attenuated virulence of ESX secretion system deficient Mtb.

Þ Thanks for your kind and insightful comments. We described ESX secretion system is essential for secretion of virulent antigen of MTB including MPT63 and MPT64 in discussion section. In this paper, we focused the function of recombinant MPT protein with interacting the host proteins in MTB challenge. So, adding the study of relation between ESX secretion system and rMPT may be hard in this paper, and we will study in further study.

9- Line 175 - 176: the effect of MPT63/64 TBK1 peptide on interaction between IRF3 and STING1 is validated in Fig 3C not Fig. 3B.

Þ Thanks for your kind comments. Right panel of Fig. 3B showed interaction between IRF3 and STING1 tagged AU1 and V5, respectively.

10- Line 331: Text says Mtb is infected intranasally. Figure legend says Mtb is infected via aerosol infection.

Þ We are sorry for our mistakes. We have corrected aerosol to intranasal in Figure 7 legend.

11- Line 394: rMPT; Line 431: Our – our

Þ We are sorry for our mistakes. We have corrected.

Reviewer 2 Report

The study done by Kim et al. reports the discovery of the Mycobacterium tuberculosis proteins-derived synthetic polypeptide that has a potential as an effective vaccine or therapeutic agent against M. tuberculosis, one of the most concerned infectious diseases worldwide with its high mortality. In this manuscript, the authors identified M. tuberculosis proteins-interacting novel host proteins, and also demonstrates that synthetic peptides based on these protein-protein interactions are useful to suppress M. tuberculosis infection. These results are definitely expected to contribute to the development of specific regulator of M. tuberculosis, and therefore are worth being published in Biomedicines. Several minor points should be addressed before final decision, which are listed below.

>Page 3, line 106–107

  • Authors wrote that “150–159 amino acids in C-terminal domain of MPT63 were important for binding with MPT63.” Is it intended to say that “MPT63 were…binding with MTP63”?

>Page 4, line 141–144

  • Authors described that “the in vitro binding between MPT64 and TBK1 or HK2 was analyzed in a binding assay using recombinant proteins with fluorescently labeled MPT63 bound with TBK1 or p47phox in high affinity (TBK1, 193nM; p47phox, 134nM)” The authors intended to mention MPT64, TBK1, and HK2?

>Page 12, line 394

  • Authors wrote that “ (5) rMP, constructed…” rMP means rMPT?

>Page 14, line 431

  • “In Our study” should be modified to “In our study”

>Fig. 2B, 2G, 2H, 3F, and S1D are not mentioned in the manuscript.

>I recommend the authors to shorten and rearrange the discussion section, as it appears too long and to contain some nonessential information.   

Author Response

Þ Thanks for your kind and excellent comments. We submit a revised version of our manuscript and a point-by-point response to the reviewers’ comments. Detailed responses are described below.

Reviewer 2
The study done by Kim et al. reports the discovery of the Mycobacterium tuberculosis proteins-derived synthetic polypeptide that has a potential as an effective vaccine or therapeutic agent against M. tuberculosis, one of the most concerned infectious diseases worldwide with its high mortality. In this manuscript, the authors identified M. tuberculosis proteins-interacting novel host proteins, and also demonstrates that synthetic peptides based on these protein-protein interactions are useful to suppress M. tuberculosis infection. These results are definitely expected to contribute to the development of specific regulator of M. tuberculosis, and therefore are worth being published in Biomedicines. Several minor points should be addressed before final decision, which are listed below.

  1. Page 3, line 106–107, Authors wrote that “150–159 amino acids in C-terminal domain of MPT63 were important for binding with MPT63.” Is it intended to say that “MPT63 were…binding with MTP63”?

Þ We are sorry for our mistakes. We have corrected (line 101-102 in Revised manuscript).

  1. Page 4, line 141–144, Authors described that “the in vitrobinding between MPT64 and TBK1 or HK2 was analyzed in a binding assay using recombinant proteins with fluorescently labeled MPT63 bound with TBK1 or p47phox in high affinity (TBK1, 193nM; p47phox, 134nM)” The authors intended to mention MPT64, TBK1, and HK2?

Þ We are sorry for our mistakes. We have revised MPT63 to MPT64 (line 134-137 in Revised manuscript).

  1. Page 12, line 394, Authors wrote that “(5) rMP, constructed…” rMP means rMPT?

Þ We are sorry for our mistakes. We have corrected (line 396 in Revised manuscript).

  1. Page 14, line 431, “In Our study” should be modified to “In our study”

Þ We are sorry for our mistakes. We have corrected (line 425 in Revised manuscript).

  1. Fig. 2B, 2G, 2H, 3F, and S1D are not mentioned in the manuscript.

Þ We are sorry for our mistakes. We have added the Fig. 2B, 2G, 2H, 3F and S1D in the Revised manuscript.

  1. I recommend the authors to shorten and rearrange the discussion section, as it appears too long and to contain some nonessential information.   

Þ Thanks for your kind and insightful comments. We have deleted the nonessential information in discussion section.

Round 2

Reviewer 1 Report

All concerned are well addressed.